# A Novel Approach against Sun Glare to Enhance Driver Safety

**Vlad-Ilie Ungureanu \*****, Răzvan-Cătălin Miclea, Adrian Korodi and Ioan Silea**

Automation and Applied Informatics Department, Politehnica University of Timisoara, 300006 Timisoara, Romania; miclea_razvan@yahoo.com (R.-C.M.); adrian.korodi@upt.ro (A.K.); ioan.silea@upt.ro (I.S.)
**\*** Correspondence: vlad95.ungureanu@gmail.com; Tel.: +40-721-346-626

**Featured Application: The presented research has the potential to be applied and used in the automotive industry.**

**Abstract:** The automotive industry is developing continuously, trying to improve, among others, the safety of drivers, passengers, and pedestrians. Using modern technology, the dangers caused by weather hazards like rain, snow, fog, or glare were identified and reduced. This paper presents an anti-glare solution using existing technologies that can be found already in a high-end car like the driver's eyes tracking systems, light intensity sensors, or head-up displays. In addition to the existing elements, a sun tracking sensor is required to detect the point where the sun light has the maximum intensity on the windshield surface. Knowing the driver's position and the point on the windshield where the sunlight has a high intensity, a dark spot can be created on the windshield in order to reduce the discomfort created by glare. Depending on the intensity of the light and taking into consideration the traffic safety laws, the spot's transparency can vary between certain limits. Therefore, the dangers caused by glare will be diminished and the risks of not observing pedestrians, other traffic participants, traffic signs, or sudden curves will be considerably lower. Another advantage of using a digital sunshade based on smart glass or in-glass transparent displays, instead of a regular sunshade, is that the whole windshield can be protected against glare not just the top of it. The results were verified and highlighted using computer simulations done via a MATLAB environment.

**Keywords:** sunshade modeling in MATLAB; solar tracking sensor; automatic anti sun-blind application; head-up display on windscreen; traffic safety

## 1. Introduction

A remarkable technological progress was done in the automotive area, all fields like comfort, safety, reliability, and production costs were improved in the past years. The current paper will focus on improving the traffic safety in difficult weather conditions. Usually meteorological phenomena like snow, high wind, fog, or heavy rain are taken into consideration and solutions for these kind of problems were proposed by other papers. For example, the study presented by [1] focuses on driving the vehicle under heavy rain weather conditions and proposes a method that helps estimating the visibility of traffic signs for the driver by using image processing. Experimental results presented in the paper showed that the proposed method improves the safety of all traffic participants.

Another study [2] shows how dangerous bad weather conditions can be for the traffic participants when combined with other elements like traffic speed, bad road quality, and road type (two-lane road, multiple-lane road, and highway). The results show how accident risks are affected by weather conditions like slippery roads, poor visibility, icy rain, or slush.

A neglected meteorological phenomenon is glare. Glare can be caused by multiple reasons, depending on the time of the day. During night, glare can be caused by the headlamps of the vehicles coming from the opposite direction. This problem can be solved by improving the technologies used for manufacturing the headlamps, as well as big companies being involved in the development of better products. Therefore, using LED and laser technologies, combined with auto-dimming technology and proper light color temperature, the glare caused by headlamps can be reduced. During day light, glare is caused by the sun that can temporarily blind the driver, thus, increasing the risk of accidents.

Head-up display (HUD) can be considered a new technology that is being used more and more these days. The research proposed by [3] presents a navigation system for motorcycles using HUD technology. It is known that a motorcycle rider moves their viewpoint in a characterful way while driving, which makes it difficult for the rider to look at small liquid crystal displays, therefore, the solution proposed is a HUD that leads to increased safety while driving a motorcycle. There are existing papers like [4] that analyze the driver's behavior while using HUD. Using a driving simulator, the driver's acceptance of the HUD location was evaluated, therefore, obtaining valuable data about where the information should be displayed and in what manner. As everywhere, there is still room for improvement regarding head-up displays and its technology, as it leads to possible issues on the driver's visual attention as stated in [5]. Therefore, shifting visual attention between virtual image and background can cause the driver to focus on the HUD and not on the road, sometimes being difficult for the driver to distinguish elements from the HUD image and the background scene.

Nowadays, cars are equipped with driver's eye tracking systems in order to monitor the loss of attention of the driver during day and night driving conditions as presented in [6], using the percentage of eye closure as an indication of the alertness level. Using this system, the car can perform several actions like breaking or turning on an alert sound in order to decrease accident risks. Furthermore, in [7] the driver's behavior in new environments was analyzed in order to determine the time spent looking at unfamiliar road signs showing how the driver tried to compensate the cognitive overload. Driving patterns are observed in paper [8], too, where gaze patterns of drivers were identified on different road turns and traffic lights. It was observed that, while taking left or right turns and waiting in traffic light, the focus of drivers considerably changes.

As of yet there are not many solutions to the glare problem. The manual sun visor is still widely used but has major drawbacks. First, it must be manually operated by the driver. The fact that the driver must operate the sun visor combined with the blindness caused by the sun can lead to an accident. As an alternative, a sun visor operated by a servo motor [9] that, based on a light sensor, decides whether to open or close the flap can be used. Second, the whole windshield is not covered by classic sun visors.

During CES 2020 [10], Bosch Virtual Visor featured a transparent LCD screen paired with a small in-cabin camera used to track the sun shining on the driver's face. The system employs artificial intelligence to locate facial features (including eyes, mouth, and nose) in order to track shadows as they move across the driver's face. A patented algorithm is then used to pinpoint where the driver's eyes are and selectively block and unblock (darken) sections of the virtual visor in real time to prevent blindness. The key benefit is that 90% of the visor always remains transparent, so the driver can still see out far more effectively than they otherwise would with a conventional fabric-covered visor. The drawback of this prototype is that the fabric-covered visor is just replaced and not fully removed.

An anti-glare prototype was proposed by [11], where the sunlight was reproduced using a projector emitting bright white light and webcams were used to track the eyes of the driver. The study shows promising results, being limited by the existing technology at the time when it was conducted. Therefore, the prototype could be tested only in a laboratory environment, hence, it was unusable in a real car test scenario.

The purpose of the current paper is to present an anti-glare system suitable for vehicles that aims to use the existing elements in the vehicle in order to increase the traffic safety. Actual technologies like eye tracking systems, HUD, and light intensity sensors are used. The only sensor that is added is

a sun tracking device. The tradeoff is that if the sun tracking sensor is added, the conventional sun visors can be removed, therefore, balancing the cost of production. The paper is based on the ideas presented in [12], focusing on the practical side of the idea, touching implementation details, and showing simulation results for the case in which the glare is caused by the direct beams of the sun. Furthermore, only the sun rays that are coming through the windshield are taken into consideration, but the presented principles can be applied for side windows as well. All the results were obtained using MATLAB simulations since the realization of a real-life prototype, usable in a car, was unattainable due to cost limitations.

## 2. Materials and Methods

The presented system relies on the following components:

- An eye tracking mechanism;
- A light intensity detection mechanism;
- A sun tracking mechanism;
- A shading mechanism.

Due to the already accepted solutions on the market for the eye tracking systems that monitor if the driver is going to fall asleep or if he is distracted, the current paper will not focus on solving this problem and will only present some existing solutions. The same applies for light intensity detection. There are systems that, depending on some thresholds, can turn on or off the car's headlights.

The components of the whole proposed digital sunshade system are briefly presented in a graphical manner in Figure 1a. The data provided by the sun tracking sensor and the camera are used to determine the location of the point on the windshield where the dark spot should be created. The opacity of the spot can be determined based on the information delivered by the light intensity sensor. The flow of the data is also highlighted by arrows. All the used components are small; therefore, they can be placed easily in a car. From a cost of production point of view, the sensors and cameras available on the market are not that expensive, due to the fact that the conventional sun visor can be removed since it no longer has a real purpose. From a hardware point of view, only the sun tracking sensor and the HUD will be presented since these two are the main components of the proposed hardware structure. For the other components, we will not indicate a specific hardware, but we will present the theoretical aspects, principles, and possible implementations in order to make the whole sunshade system work.

Figure 1b presents a block diagram describing the closed feedback loop system with its main components. The input variables, functional correlations, and how the new position of the center of the dark spot is created are also highlighted. The proposed cyclicity of the system is 1 s, therefore, each second, the sensors will measure the sun's position and light intensity. The provided information, alongside predefined limitations, will serve as input variables for the phase in which the center of the dark spot is computed. Having located the new point, the decision to move the whole darker area on the windshield is taken if a threshold of 1 cm is exceeded. Meaning that if the distance between the previous center and the new one exceeds the defined threshold, new instructions will be generated and provided to the driver, taking into consideration the predefined shape and size of the dark area. From this point on, the instructions will be forwarded to the smart glass and the digital sun visor will be created. Most likely, the center of the area will be the darkest and the transparency will increase towards the limits. The coordinates computed previously will serve as feedback and the whole process will repeat as long as the digital sun visor is needed.

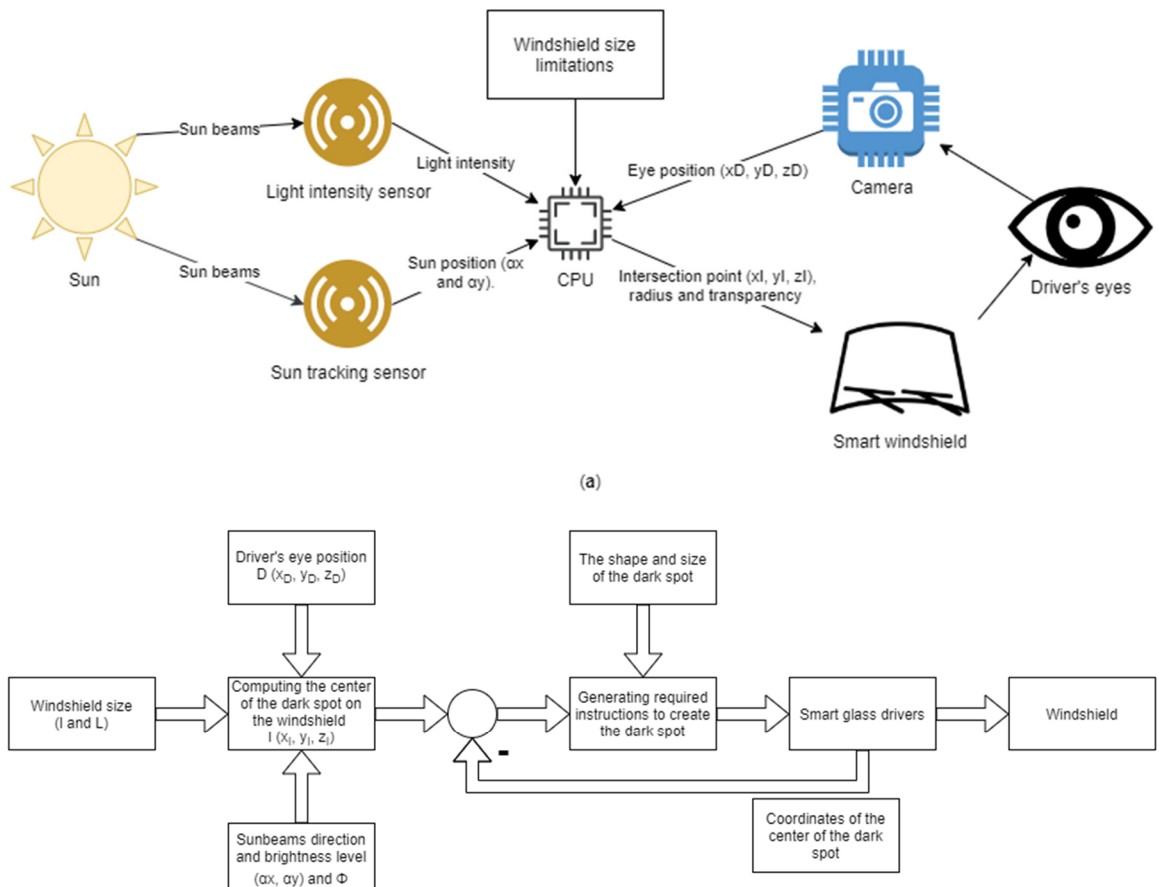

(a)

**Figure 1.** (**a**) A graphical overview of the digital sunshade system. (**b**) Block diagram of the digital sunshade system.

## 2.1. Hardware Structure

In the following subchapter, we will present some existing hardware components on the market that can be used to achieve the hardware part of the system.

The proposed solar tracking sensor is made by Solar MEMS Technologies and is highlighted in Figure 2.

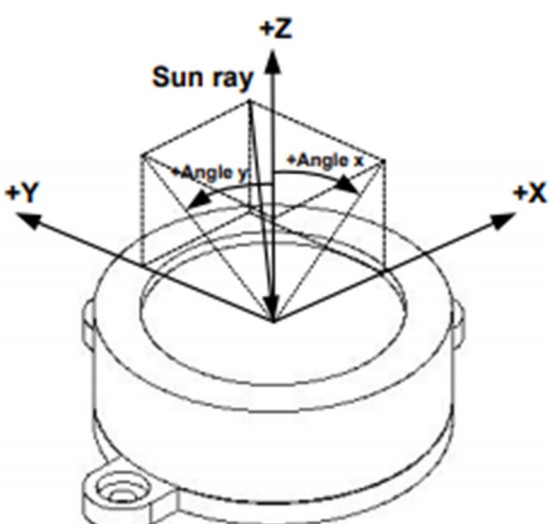

**Figure 2.** ISS-DX sensor. Reference for measured angles.

The ISS-D60 sensor has the following main features [13], briefly presented in the Table 1.

**Table 1.** ISS-D60 features.

| Parameter | Value | Unit | Comments |
|---|---|---|---|
| Sensor type | 2 axes | - | Orthogonal |
| Field of view | 120 | ° | Aperture of the cone view |
| Accuracy | <0.4 | ° | - |
| Precision | <0.06 | ° | Sensitivity |
| Angle resolution | 0.01 | ° | - |
| Supply voltage | 5 to 12 | V | - |
| Average power consumption | 33 | mA | - |
| Operating temperature | −40 to +85 | °C | Industrial temperature range |
| Diameter | 80 | mm | - |
| Height | 27 | mm | - |
| Level of protection | IP65 | - | CEI 60529 Standard |
| Expected lifetime | >10 | years | - |

The ISS-DX sensor can be used for multiple applications like sun tracking systems, heliostats, altitude control using light sources, or navigation systems. In [14], the sensor was used in a navigation system designed for micro planetary rovers. Therefore, an even cheaper sensor can be used since this one can be considered overqualified for this task.

The head-up displays evolved a lot in the last years and it is a domain in which big automotive companies invest. Next, we will present a series of alternatives that can be used. Among the solutions, we can mention smart glass, which is a type of glass whose opacity can be controlled by applying a voltage, heat, or light source. In this manner, the glass changes from transparent to translucent, in consequence, blocking a part of the light wavelengths. At CES2016 [15] and CES2017 [16] Continental Automotive presented this product which, based on light sensors, can be tinted at exactly the point where the sun is shining on (Figure 3).

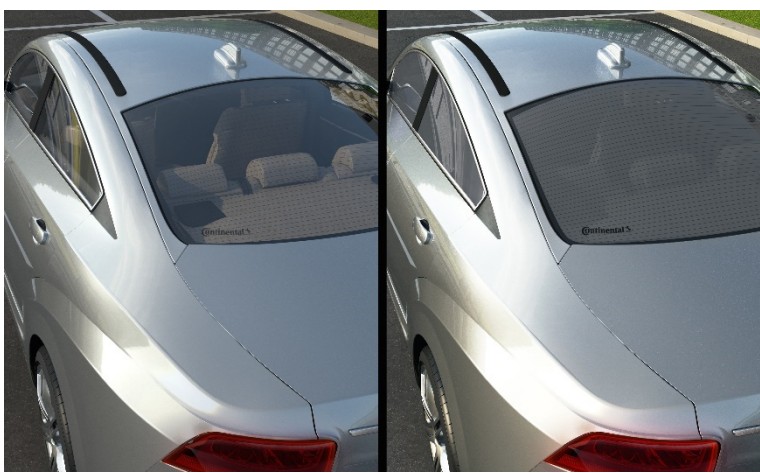

**Figure 3.** Smart glass concept.

Another extraordinary piece of technology is built by Lumineq [17]: in-glass displays which are perfect for this kind of application. The displays offer 80% transparency for segment displays and 70% transparency for matrix and a 360-degree viewing angle. The product is marketed as ultra-reliable, having an increased tolerance to shock, vibration, and extreme weather. Moreover, the type, size, shape, and the place where to insert the display is fully customizable. An example of an in-glass display is presented in Figure 4.

Taking a more futuristic approach, Alticast Corp [18] presented a demo regarding a future transparent flexible display for car information, HUD, and driving automation. The demo used a TFD made by LG (Figure 5).

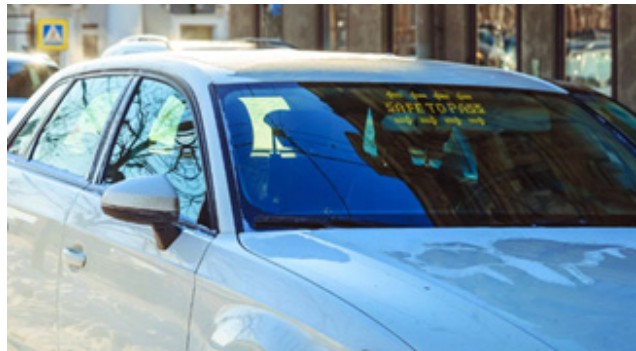

**Figure 4.** In-glass display concept.

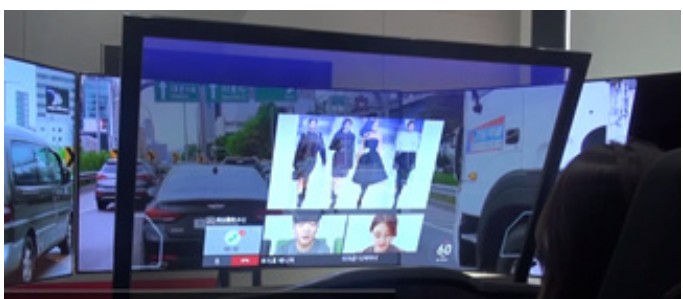

**Figure 5.** Smart windshield concept.

*2.2. Theoretical Aspects*

Having all the information that is needed, we can proceed to determine where on the windshield the darker spot shall be created. The solution is obtained by transposing the system to a 3D geometry problem and then solving it. In Figure 6, the geometrical overview of the system can be observed.

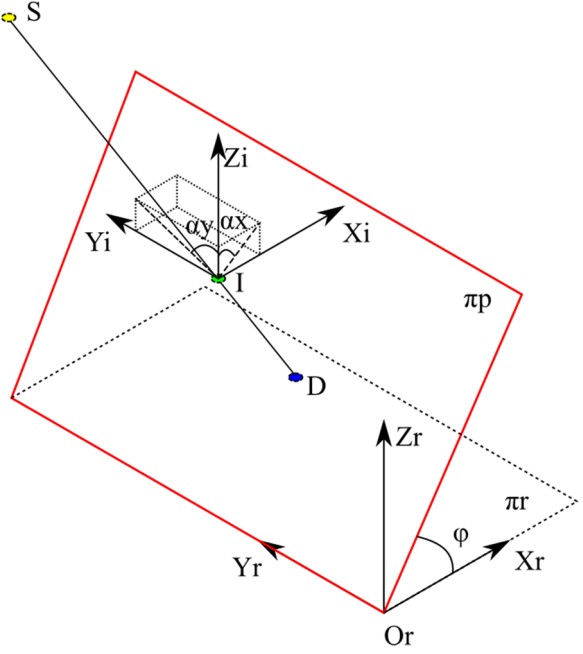

**Figure 6.** Geometrical overview of the system.

The reference plan is noted with πr, *Or* being the origin point, more precisely, the point where the sun tracking sensor is placed inside the car (the bottom-left corner of the windshield). The windshield plan is marked using πp, and the angle between the reference plan and the windshield plan is marked by $\varphi$. The information about the driver's position can be obtained from the camera-based tracking system, therefore, D represents the driver's eye. The sun is marked using S, the intersection between the sunbeam and the windshield is noted as *I*. The sun tracking sensor returns the angles $\alpha x$ and $\alpha y$. The reference system's (in πr plan) axes are parallel with the *Xi I Yi* system.

The purpose is to determine the coordinates of *I* as the intersection point between the sunbeam and the windshield. The coordinates will depend on D $(x_D, y_D, z_D)$, $\varphi$, $\alpha x$, and $\alpha y$. All of these being known variables.

In order to simplify the problem, we split it in multiple parts. First, we need to determine the equation for the plans πr and πp. To be noted that $\vec{i}$, $\vec{j}$, and $\vec{k}$ are unit vectors (each corresponding to *OrXr*, *OrYr*, and *OrZr*), besides these vectors, there are other vectors marked by an arrow above and used in the following equations. These mentioned vectors were not added to the figures because it would decrease the readability, making the images hard to visualize due to high complexity. From Figure 7 we can obtain plan's πr Equation (1) and observe that the plan πp can be determined by *OrYr*, *OrP*, and *Or* (two lines and one point).

$$\pi r : z = 0, \tag{1}$$

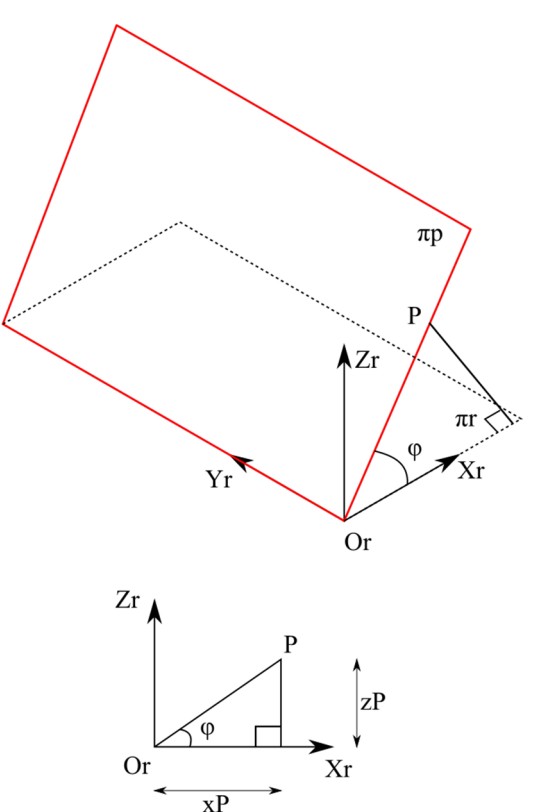

**Figure 7.** Support for determining the equations for πr and πp.

The equation describing *OrYr* is shown in Equation (2). Since *Or* is the reference point it has the following coordinates: $(0, 0, 0)$. Inserting Equation (3) in Equation (4), the equation for *OrP* is obtained (see Equation (5)).

$$OrYr : \frac{x}{0} = \frac{y}{1} = \frac{z}{0} \rightarrow \vec{v_{OrYr}} = \vec{j}, \tag{2}$$

$$\tan\varphi = \frac{zP}{xP} \rightarrow zP = xP * \tan\varphi, \tag{3}$$

$$OrP: \frac{x}{xP} = \frac{y}{0} = \frac{z}{zP} \rightarrow \frac{x}{xP} = \frac{y}{0} = \frac{z}{xP * \tan\varphi}, \tag{4}$$

$$OrP: \frac{x}{1} = \frac{y}{0} = \frac{z}{\tan\varphi} \rightarrow \vec{v_{OrP}} = \vec{i} + \tan\varphi\,\vec{k}. \tag{5}$$

Then, using known formulas, such as Equation (6), we can obtain the equation for $\pi$p (see Equation (7)).

$$\pi\mathrm{p}: \begin{vmatrix} x - x_{Or} & y - y_{Or} & z - z_{Or} \\ u_i & u_j & u_k \\ v_i & v_j & v_k \end{vmatrix} = 0, \tag{6}$$

$$\pi\mathrm{p}: \begin{vmatrix} x & y & z \\ 0 & 1 & 0 \\ 1 & 0 & \tan\varphi \end{vmatrix} = 0 \rightarrow \tan\varphi * x - z = 0. \tag{7}$$

The next step is to determine the equation describing *SD*. Figure 8 will serve as support in order to solve this problem.

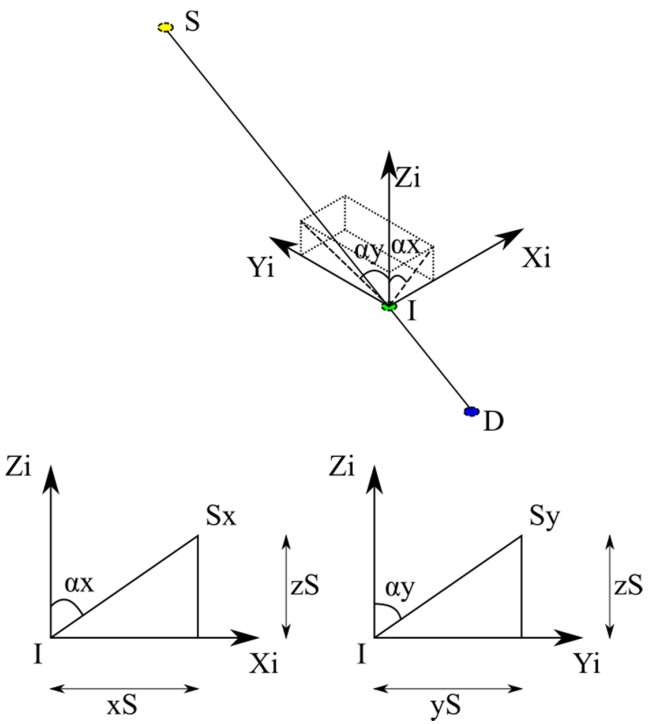

**Figure 8.** Support for determining equation for SD.

We can see that *SD* can be determined by $\vec{v_{SD}}$ and D ($x_D$, $y_D$, $z_D$). Inserting Equations (8)–(10), we obtain the desired Equation (11). Now all that is left is to determine the coordinates of the intersection point *I*, knowing Equation (12).

$$\tan\alpha x = \frac{xS}{zS} \rightarrow xS = zS * \tan\alpha x, \tag{8}$$

$$\tan\alpha y = \frac{yS}{zS} \rightarrow yS = zS * \tan\alpha y, \tag{9}$$

$$SD: \frac{x - x_D}{xS} = \frac{y - y_D}{yS} = \frac{z - z_D}{zS}, \tag{10}$$

$$SD: \frac{x - x_D}{\tan\alpha x} = \frac{y - y_D}{\tan\alpha y} = \frac{z - z_D}{1}. \tag{11}$$

$$I = SD \cap \pi\mathrm{p} \tag{12}$$

The coordinates of *I* point are obtained by solving the system of equations presented in Equation (13).

$$\begin{cases} \tan\varphi * x_I - z_I = 0 \\ (x_I - x_D) * \tan\alpha y - (y_I - y_D) * \tan\alpha x = 0 \\ x_I - x_D - (z_I - z_D) * \tan\alpha x = 0 \\ y_I - y_D - (z_I - z_D) * \tan\alpha y = 0 \end{cases}. \tag{13}$$

The equations system 13 is extended by adding limitations caused by the fact that the windshield has a fixed size, therefore, the point *I* can be discarded if it is not on the windshield area.

### 2.3. Implementation Aspects

Since we did not have all the necessary resources to create a real car prototype, a simulation was preferred in order to test and confirm the theoretical part. It is known that not all real-life issues can be covered by simulations, therefore, we could only predict some aspects like the runtime, task prioritization, or memory usage. Moreover, it is considered that glare represents an issue only on sunny weather conditions or when the driver provides a feedback by pressing a button. Light intensity sensors and sun tracking sensor, alongside some extra data that can be received from the car like the current time of the day, can be used to determine if the dark spot will be created or not. Several scripts were created to cover the use cases when the driver position is fixed, and the sun's position is mobile and vice versa. All variables are configurable in an input script, and in this manner, the user can control the following aspects of the simulation: the size of the windshield (L = width and l = height), the angle between the reference plan and the windshield plan ($\varphi$), the coordinates of the driver's eye (D), and the sun's position ($\alpha$x and $\alpha$y). The flow chart of the whole sun shading modeling process is described in Figure 9.

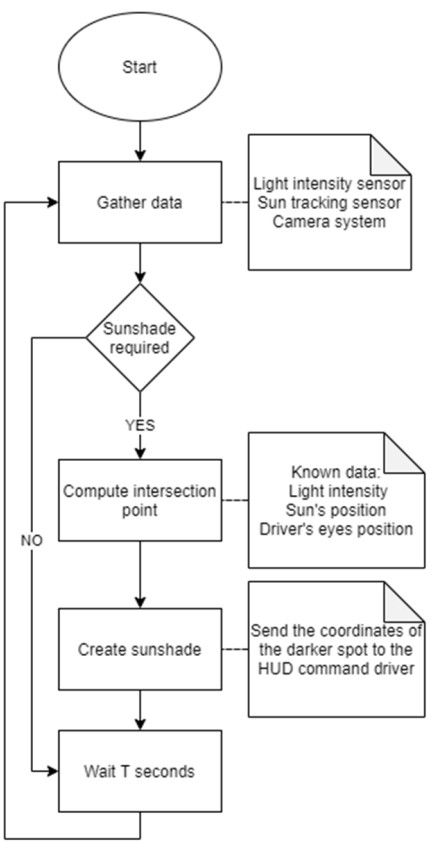

**Figure 9.** Shading system flow chart.

The components involved in providing the information needed to operate the system are a set of photo cameras tracking the position of the driver's eyes, the sensor that provides the direction of the sun's rays, and the sensor that provides the level of brightness. If the digital sunshade is needed, all the data are sent to the ECU. The computational power required should be present on all the vehicles, due to the fact that this is not a high priority task and its cyclicality can be configured to a value like $T = 1$ s. The priority and cyclicality are based on the fact that even if it is a road safety feature, it cannot be compared to other safety features such as automatic breaking or airbag triggering. The variations in this process are slow since the direction of the car is not changing that often even in the city, the same hypothesis being applied for the sun or driver's head movement. After the intersection point between the sunbeam and the windshield is found, the HUD driver is instructed in concordance and the dark spot is created, its transparency being in concordance with the light intensity.

As mentioned above, the current paper does not aim to solve the driver's eye tracking problem. Therefore, some already existing solutions will be presented as examples as what could be used to cover this part of the whole environment. A functioning camera-based eye tracking system is presented in [19]. Some very important aspects are that the system is capable of having the following features at real-time speed: face tracking, pose estimation, gaze direction estimation, and failure detection and recovery. The hardware structure of this kind of system is presented in Figure 10.

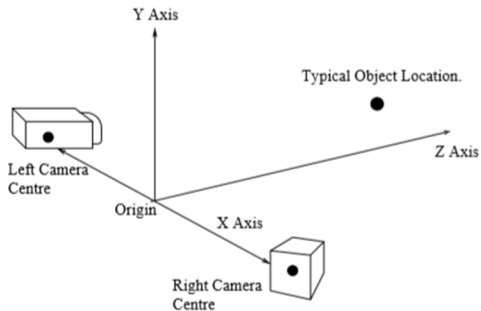

**Figure 10.** Eye tracking system based on two cameras.

As can be seen, the two cameras are placed equidistant from the origin point and are rotated towards the origin in the horizontal plane. Specific regions of the human face are tracked like the corners of eyes, mouth, nose, ears, and eyebrows. Next, the eyes are considered as spheres and the gaze direction is given by the direction of a line starting from the eyeball center. The intersection point of both eyes is the gaze point. Due to the progress made in the last years on the image acquisition, such a system would not add much costs and will provide very good results.

Nowadays, even more companies like Smart Eye [20] offer solutions based on a single camera and LED lights (Figure 11). The infrared light's purpose is to reflect from the eye's cornea, and thus, the camera will be able to detect the eye's position. From this point, image processing is used to track the driver's gaze and eye position. Furthermore, algorithms based on AI (artificial intelligence) principles are used to interpret the gestures of the driver.

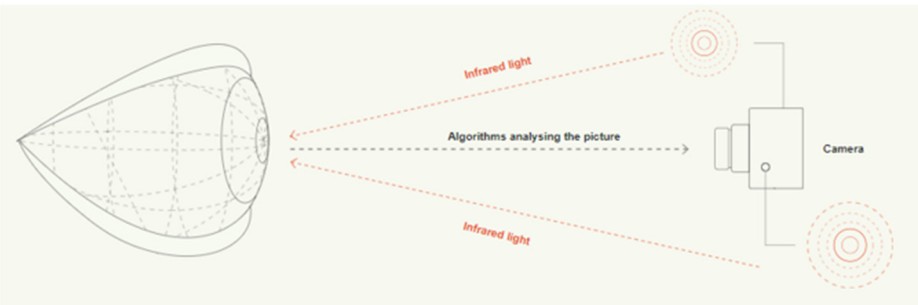

**Figure 11.** Eye tracking system based on a camera and LED lights.

Products like Smart Eye Pro DX [20] can be easily used in a vehicle being able to detect head position, eye position, and identify different aspects like if the driver is sleepy or not.

A critical point can be the computational power required in order to solve the system of equations presented in 13 that will be extended with the following limitations 14 in order to assure the fact that the intersection point will be limited to the size of the windshield or, after all, limited to any size (top part of the windshield, for example).

$$\begin{cases} y_I \geq 0 \\ y_I \leq L \\ z_I \geq 0 \\ z_I \leq \sin\varphi * l \end{cases}. \tag{14}$$

All the simulations were run on a PC with the following specifications: Intel Core i7-8700 CPU 3.20 GHz (12 CPUs) processor, 16 GB of RAM memory, and 480 GB SSD. Using the MATLAB built-in measuring feature, solving the system of equations took 119 milliseconds. It is to be noted that Windows OS and MATLAB were responsible for used algorithms, prioritization, etc. In order to have a better appreciation of the time required to solve the equations in a real environment, a less powerful board was chosen: ATmega328P microcontroller with 32 KB of flash memory, 2 KB SRAM, and 16 MHz clock speed. On this microcontroller, the time required to solve the equations, using Equation (15), was under 1 millisecond (0.8 microseconds). When we used Equation (16), the time required to solve the equations was around 1.2 milliseconds. On the system, the only process that was running was the algorithm that was solving the equations. Having this information, it is clear that the process can run on a real system in a low priority task, without interfering with higher priority sequences. From the memory usage perspective Equation (15) comes, again, with better performances since fewer variables are used.

In order to solve the system of equations from 13, the coordinates of the intersection point are computed using Equation (15). From the first 3 equations, the solution can be obtained and the 4th equation is used just to check if the solution is valid or not. This solution was reached by solving the system of equations by hand.

$$\begin{cases} x_I = \frac{x_D - \tan\alpha x * z_D}{1 - \tan\alpha x * \tan\varphi} \\ y_I = \frac{\tan\alpha y(x_D - \tan\alpha x * z_D)}{\tan\alpha x \, (1 - \tan\alpha x * \tan\varphi)} + \qquad y_D - \frac{\tan\alpha y * x_D}{\tan\alpha x}. \\ z_I = \tan\varphi * x_I \\ y_I - y_D - (z_I - z_D) * \tan\alpha y = 0 \end{cases} \tag{15}$$

Another way to achieve the solution can be by using Cramer's rule to solve the first 3 equations, and then the 4th equation is used just to check if the solution is valid or not. The coefficients needed to calculate the determinants are presented in Equation (16) and were obtained from Equation (13).

$$\begin{vmatrix} \tan\varphi & 0 & -1 & 0 \\ \tan\alpha y & -\tan\alpha x & 0 & \tan\alpha y * x_D - \tan\alpha x * y_D \\ 1 & 0 & -\tan\alpha x & x_D - \tan\alpha x * z_D \end{vmatrix}. \tag{16}$$

For both variants, limitations presented in Equation (14) shall be used to limit the intersection point to the windshield size.

## 3. Results

In this chapter the results achieved will be presented and discussed. All the tests were done via MATLAB simulations. There are several scenarios that were created in order to test and highlight the principles described in the previous chapters.

In Figure 12, the simulation flow chart is presented. The user shall configure the input data in a separate file then it can run the simulation. For readability, a simulation file for each use case was created. After the data were loaded into simulation, based on the data provided by the sun tracking

sensor, the coordinates of S point were computed. Using Equation (11) and imposing some limitation like the distance between D and S or the fact that S must be on the other side of the windshield compared to D, the position of the sun is obtained. This step is used mostly for data visualization. In this way, the DS line can be plotted. Next, the intersection point was computed based on Equations (13) and (14). The user can provide input vectors of values for the sun's position ($\alpha x$ and $\alpha y$) or for the coordinates of the driver's eye, D ($x_D$, $y_D$, $z_D$). After all the input data are processed and all the intersection points are available, the results are plotted in order to provide a good data visualization. All the computed data are also provided in numerical form if needed for an interpretation.

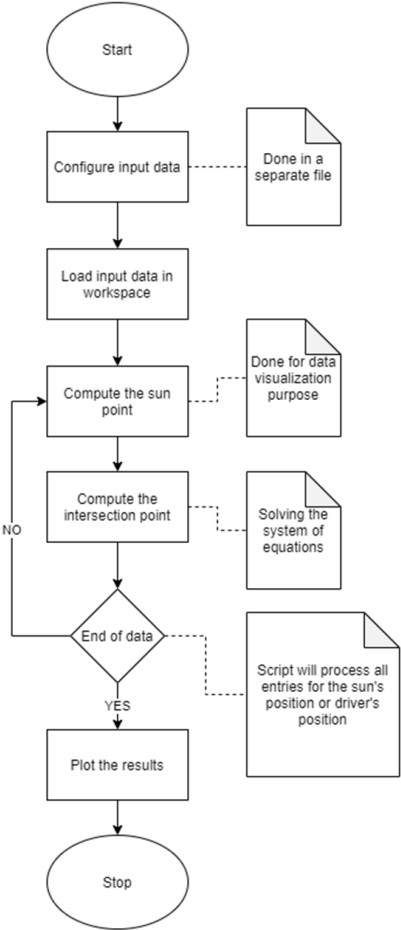

**Figure 12.** Simulation flow chart.

The first test scenario is a simple one. The sun S and the driver D are considered as fixed points. As can be seen in Figure 13, an intersection point I was obtained. Therefore, the center of the dark spot was to be created there. The line between DS and ID is marked in purple, the plane is colored blue, and the intersection point I is marked with a green cross.

For the next test scenario, the driver was considered a fixed point and the sun was mobile. A limitation that the points of interest are only on the top part of the windshield was applied. The practical use of this limitation is that in some cases the whole windshield is not covered by a HUD. Therefore, the points of interest were only the ones that were inside the display area. In Figure 14, the results of the simulation can be observed.

The next test scenario covers the part where the driver is moving as well. Therefore, the sun was considered a fixed point and the driver is mobile. The driver's movement was described by a sphere with a fixed radius. The same limitations were applied as in the previous test. In Figure 15, the results of the simulation can be observed.

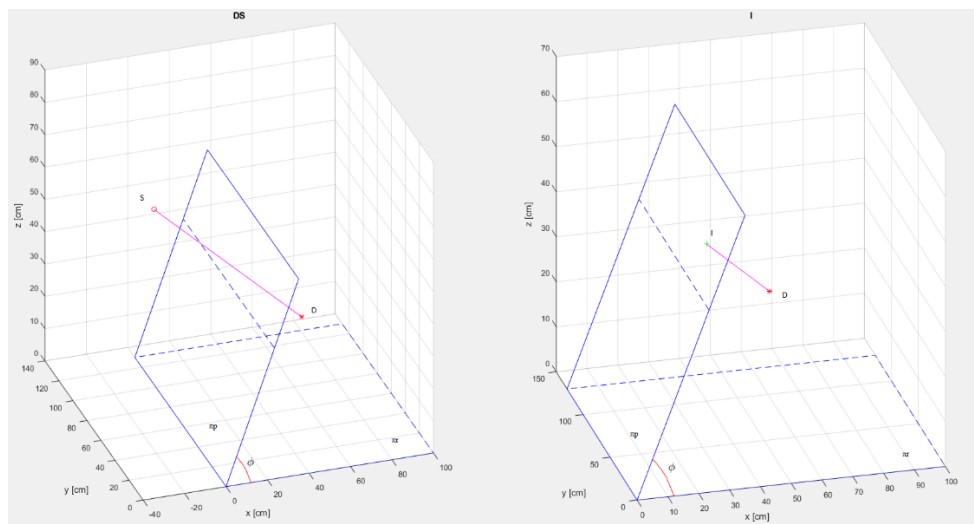

**Figure 13.** Test scenario 1—sun (S) and driver's eye (D) are fixed points.

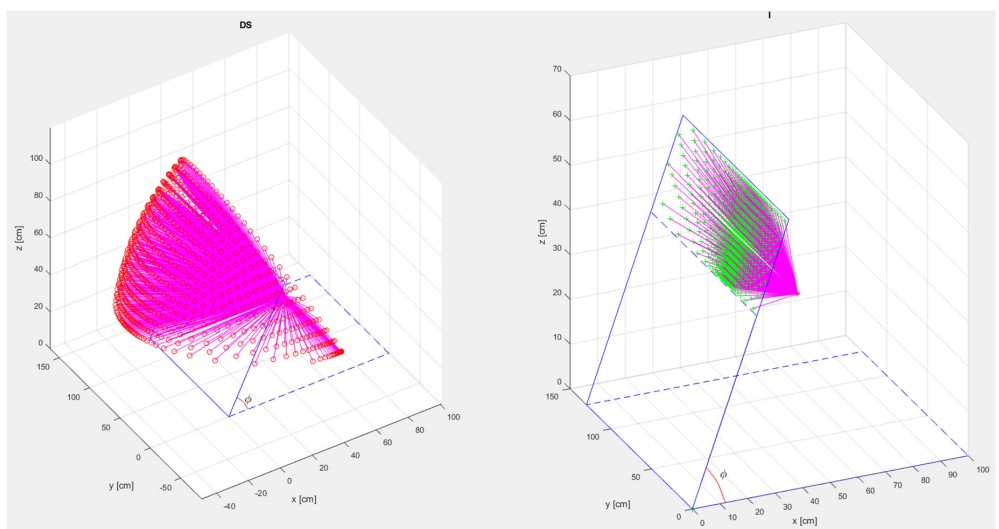

**Figure 14.** Test scenario 2—S is a mobile point and D is a fixed point.

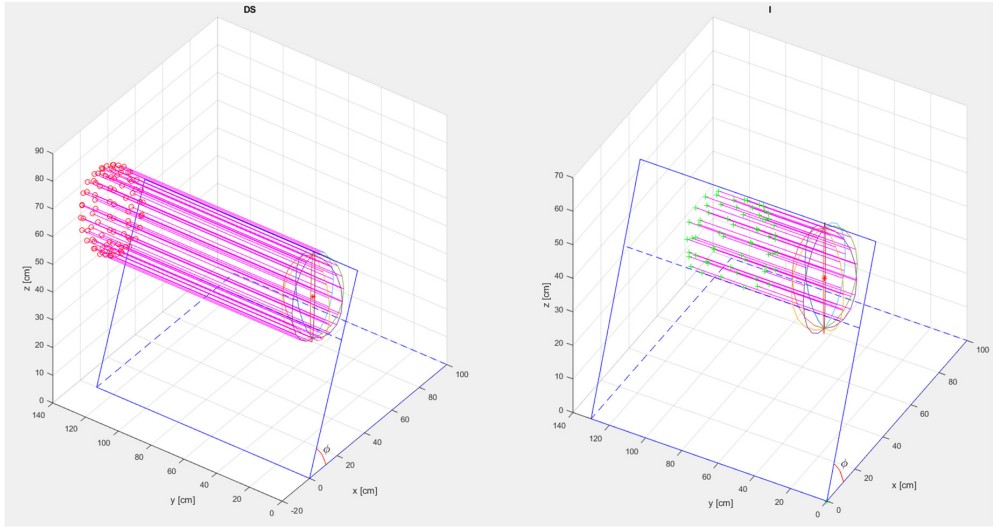

**Figure 15.** Test scenario 3—S is a fixed point and D is a mobile point.

For the final test scenario, the sun's position was changing, and the driver's eyes were moving as well. The driver's movement were described by a sphere with a fixed radius. The same limitations were applied as in the previous test. In Figure 16, the results of the simulation can be observed.

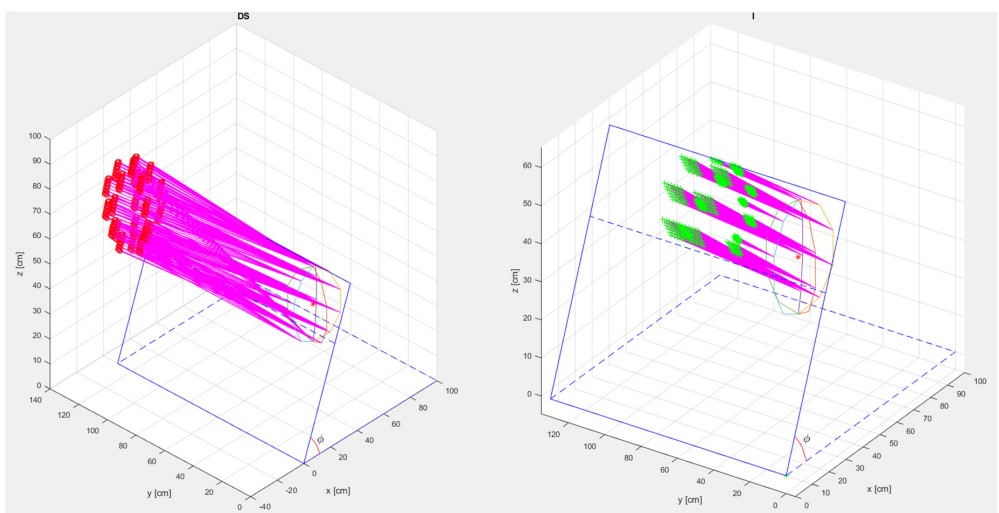

**Figure 16.** Test scenario 4—S and D are mobile points.

## 4. Discussion

In this paper we presented a study on the applicability and the benefits of an anti-glare system. Several test scenarios were probed using multiple MATLAB simulations in order to confirm the usability of such a system. Considering the progress registered in the HUD field in the last years, it seems only a matter of a few years until such a system will be realized at a cost of production that is low enough to bring benefits for both the driver and the manufacturer. Indeed, the classic sun visor is outdated compared to the other components of a modern vehicle and is not contributing to the safety of the driver as much as a digital sun visor.

The presented project can be improved on different levels, starting with the hardware parts. The proposed sun tracking sensor is overqualified for a project like this since a sensor with lower precision, resolution, and accuracy would still work fine. In this manner, the costs would be reduced. An improvement from a costs point of view would be to remove the sun tracking sensor and try to obtain all the information required just by processing images.

Another potential update is that the glare problem is approached only for direct sunbeams. As is known, there are some scenarios in which the sunbeams reflected by the road are harmful, too. In this case, we consider that the same handling shall be applied, using a HUD and creating a dark spot where the reflected sunbeams are producing discomfort. The analysis of this specific case and the calculation of the dark spot's position on the windshield are not presented in this paper.

In the future, the simulation environment shall be replaced with a real prototype that can be tested in a laboratory, and then on a real-life driving test scenario.

The merit of the project consists in the fact that a large variety of knowledge from different domains was used and combined in order to propose a working product that will increase the safety of all traffic participants. The tests done prove the value and the applicability of the ideas, algorithms, and principles implemented in the paper. The presented research represents a starting point for those who develop anti-glare systems and provides good understanding of this topic. Apart from the industrial benefits, this paper can be used in didactic activities as well, showing students how to apply theoretical knowledge in order to solve a real-life problem.

**Author Contributions:** Ideas: R.-C.M. and V.-I.U.; concept, methodology, and development: V.-I.U., I.S., and R.-C.M.; software implementation: V.-I.U.; analysis of results and discussions: I.S., V.-I.U., R.-C.M., and A.K.;

writing scientific paper: V.-I.U.; supervision, validation, and financing: I.S. and A.K. All authors have read and agreed to the published version of the manuscript.

**Funding:** This research received no external funding.

**Conflicts of Interest:** The authors declare no conflict of interest.

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
