# Peer review of "A Novel Approach against Sun Glare to Enhance Driver Safety"

_applsci, doi:10.3390/app10093032_

Round 1

Reviewer 1 Report

The article is on an interesting topic, such as trying to solve with intelligent systems the safety problem caused by glare. The article is generally well written and structured. Below are my comments.

One major comment is that, being only a simulation rather than an on-road study, the time lag between the different stages of the experiment cannot be observed and that there could be different issues in the application of such sensor in the real world (e.g., weather, obstacles, etc.). Please discuss this point.

I have also some additional minor comments:

  • The format of references in the text is sometimes ankward (e.g., "the paper [X]", the journal [Y]"). And in the references section as well.
  • I see some unnecessary emphasis in describing commercial articles (i.e., lines 130-156).
  • Language should be revised in some points (e.g., only one example: "can be obtain" instead of "can be obtained" at line 162).
  • When recalling Equation, it should be written something like: see Eq. X, or y (putting only the number of the equation in parenthesis may be confused with literature studies).

Author Response

Thank you for the review. The comments were very helpful to us in improving the quality of our manuscript.

Please see the attachment for our point-by-point response.

Author Response

(The authors gave the same response as above.)
